# Arteriovenous Malformation Hemorrhage in Pregnancy: A Systematic Review and Meta-Analysis

**DOI:** 10.3390/ijerph192013183

**Published:** 2022-10-13

**Authors:** Ruhana Che Yusof, Mohd Noor Norhayati, Yacob Mohd Azman

**Affiliations:** 1Department of Family Medicine, School of Medical Sciences, Universiti Sains Malaysia, Kubang Kerian 16150, Malaysia; 2Medical Development Division, Ministry of Health, Level 7, Block E1, Parcel E, Federal Government Administrative Centre, Putrajaya 62590, Malaysia

**Keywords:** arteriovenous malformation, pregnancy, pooled proportion, double arcsine transformation, rare disease

## Abstract

Hemorrhage of arteriovenous malformation (AVM) is a rare condition during pregnancy. This study was proposed to pool the proportion of AVM hemorrhage per pregnancy. A systematic review and meta-analysis with three databases were performed to review the studies published until April 2022. The Newcastle Ottawa Scale was used for risk assessment of data quality. The meta-analysis was conducted by a generic inverse variance of double arcsine transformation with a random model using Stata software. Twelve studies were included in this review. The pooled proportion of AVM hemorrhage per pregnancy was 0.16 (95% CI: 0.08, 0.26). The subgroup analyses were carried out based on world regions and study designs, and the study duration with the highest proportion of each subgroup was Europe [0.35 (95% CI: 0.02, 0.79)], with retrospective review [0.18 (95% CI: 007, 0.32)] and 10 to 20 years of study duration [0.37 (95% CI: 0.06, 0.77)]. The AVM hemorrhage per pregnancy in this review was considered low. However, the conclusion must be carefully interpreted since this review had a small study limitation.

## 1. Introduction

Arteriovenous malformation (AVM) is an abnormal vascular lesion that could be a tangle of vessels of various sizes wherein there are one or greater direct connections among the arterial and venous circulations [1]. It also deprives the surrounding tissues of blood supply and nutrients, produces venous hypertension and localized edema and overloading the heart may cause congestive cardiac failure [2]. Although brain AVMs are frequently assumed to be congenital, there is no concrete proof that they develop in pregnancy. Only a tiny percentage of AVMs present at or shortly after birth, and the majority do and most likely grow and advance in later years of life [3]. AVMs can theoretically grow in every organ, but the most frequently affected organs are the lungs, brain and liver. These AVMs are susceptible to rupture and hemorrhage, leading to major morbidity and mortality [4].

Arteriovenous malformation is listed in the National Organization for Rare Disorders database as a rare disease. Based on available population data in the United States, an estimated 3000 to 30,000 people are diagnosed with an intracranial AVM [5], and the prevalence is 1 to 9 per 100,000 population [6]. Population estimations for other types of AVM are currently not available.

In pregnancy, the presentation of AVM is usually a result of a hemorrhage following a rupture [7]. Whether pregnancy is a risk factor for hemorrhage from AVMs is not well known and controversial [7]. However, hemodynamic changes during pregnancy significantly increase the risk of hemorrhage from AVMs [8]. An earlier study [9] showed that AVM carries an 87% risk of hemorrhage, with poor outcomes for the baby in a subsequent pregnancy if the AVM is untreated. However, a more recent study [10] found that the risk of the first hemorrhage for pregnant women with an unruptured AVM was only 3.5%. It is similar to the known annual bleeding rate in the non-gravid population with an unruptured AVM.

Apart from AVM morphology, other factors that increase bleeding from AVMs during pregnancy are younger age (20–25 years) and primigravida [11]. Once hemorrhage occurs, it accounts for 5–12% of all maternal deaths and remains the third most common non-obstetric cause of maternal morbidity [10]. During pregnancy, AVMs may present with severe headaches, meningism and photophobia and can be confused with eclampsia [2]. The diagnosis is confirmed by computed tomography or lumbar puncture and cerebral angiography [2]. Pregnancy with AVM needs special care depending on the maternal consciousness level, hemorrhage site and pregnancy stage [12].

Since AVM hemorrhage during pregnancy is a rare condition, this study was conducted to pool the proportion of AVM hemorrhage per pregnancy to provide a current estimation of AVM hemorrhage during pregnancy.

## 2. Materials and Methods

### 2.1. Types of Studies

The observation of AVM hemorrhage during pregnancy was evaluated by systematic review and meta-analysis. The outcome measured was AVM hemorrhage per pregnancy. The Preferred Reporting Items for Systematic Reviews and Meta-Analyses (PRISMA) 2020 guidelines [13] were used as a guide to review the studies and were registered in PROSPERO (CRD42022342799).

### 2.2. Search Methods

A search was conducted systematically to identify relevant articles. Databases involved were MEDLINE (PubMed), EBSCO and Scopus based on the search terms “(((risk) AND ((arteriovenous malformation)) OR (AVM)) AND (rupture)) OR (hemorrhage) AND (pregnancy)”. All studies published up until April 2022 were screened for their eligibility to be included in this review. The search was limited to full-text articles written in the English language only due to language barrier limitations. Secondary citations from the articles were cross-checked to locate other potentially acceptable research.

### 2.3. Study Selection

All the records found by the search approach were exported to EndNote software. Duplicate articles were removed. The irrelevant studies were removed by an automation tool using the terms (review) or (protocol) or (qualitative) or (“meta-analysis”) or (“case report”). Two independent reviewers checked the titles and abstracts of the identified papers, and the full texts of eligible papers were obtained and thoroughly examined to assess their eligibility. A third reviewer was consulted in the event of a conflict between the two reviewers. The search approach was depicted in the PRISMA flow chart, which included and excluded studies as well as the grounds for exclusion.

### 2.4. Data Extraction and Management

The extracted data were entered into Microsoft Excel (Microsoft Office Professional Plus 2016). The data included the first author, year of publication, study location, study design, study duration, study population, sample size, AVM location, outcome measures and data to generate effect estimates if applicable. The outcomes of interest were the number of pregnancies and AVM hemorrhages during pregnancy. Studies with incomplete data were excluded from the review.

### 2.5. Assessment of Risk of Bias

Assessment of the risk of bias for data quality was conducted using the Newcastle Ottawa Scale (NOS) for cohort studies [14]. The NOS is divided into three domains (selection, comparability and outcomes) with a total of eight specific items. Every item on the scale is scored from one star, except comparability, which can be tailored to the particular topic of interest to score up to two stars. Based on ref. [15], a “good” quality score requires 3 or 4 stars for the selected domain, 1 or 2 stars in the comparability domain and 2 or 3 stars in the outcomes domain. A “fair” quality score needs 2 stars in the selection domain, 1 or 2 stars in the comparability domain and 2 or 3 stars in the outcomes domain. Meanwhile, a “poor” quality score demands 0 or 1 star in the selection domain, 0 stars in the comparability domain or 0 or 1 star in the outcomes domain. Two authors reviewed the studies and assessed the bias independently.

### 2.6. Measures of Outcome

The outcome was measured as a proportion of the total number of AVM hemorrhages during pregnancy over the total number of pregnancies. However, since AVM hemorrhage is a rare event and the sample size was small, a transformation of the proportion by the double-arcsine method that employed the variance stabilizing transformation [16] was applied. Transformations to the data were applied in order to make the distribution of the data normal as much as possible to enhance the validity of the statistical analyses [17]. After transforming the observed proportions, all analyses were conducted with the transformed proportion as the effect size statistic and the inverse of the transformed proportion as the study weight. For reporting, the transformed summary proportion and its confidence interval were converted back to original proportions for interpretation [18].

### 2.7. Data Synthesis

The analysis was performed by Stata 13.1 software (StataCorp, Texas) for the double-arcsine transformation. Effect sizes by proportion, standard error and sample weight were calculated. A generic inverse variance with a random-effects model by the Der Simonian and Laird method was used to pool the proportion. Then, the transformed summary proportion was converted back to the original proportion. The I^2^ statistic was used to assess heterogeneity and the guide was used as outlined: 0% to 40% might not be important, 30% to 60% may represent moderate heterogeneity, 50% to 90% may represent substantial heterogeneity and 75% to 100% would be considerable heterogeneity [19]. If there was a possibility of publication bias, visualize assessment of funnel plots and Egger’s test statistics were used. Sensitivity analyses using the leave-one-out analysis were conducted to assess the influential outliers in the synthesized results.

A subgroup analysis was performed based on regions (Europe, America and Asia), study designs (retrospective review and case-cross-over) and study duration (less than 10 years, 10 to 20 years and more than 20 years).

## 3. Results

### 3.1. Study Selection

A total of 561 studies were identified through the primary search databases (PubMed, EBSCO and Scopus) and secondary citations. Three hundred and fifty studies were removed due to duplication and were marked as ineligible by automation tools. Eighty-three studies, including seven studies via citation searches, were screened and retrieved, and their eligibility was determined. After the assessment, 12 studies met the inclusion and exclusion criteria (Figure 1).

### 3.2. Study Characteristics

The total number of patients involved in this study was 2578, with a total number of pregnancies of 1,962 and a total number of hemorrhages during pregnancy of 158. The included studies were published from 1974 to 2021 in three regions: Europe, which involved the United Kingdom [9,20] and Scotland [21]; America, which involved the United States [10,22,23,24] and North America [25]; and Asia, which involved China [26,27,28] and Japan [12].

These studies applied three types of study design: retrospective review [9,10,12,20,22,23,24,25,27,28], case-cross-over design [26] and a combination of self-controlled case-series and case-crossover design [21]. The study with the combination of case series and case-crossover designs was treated as a case-crossover design in the meta-analysis.

Since AVM hemorrhage is a rare disease event, these studies take a longer duration. The study duration ranged from five years to 51 years and was grouped into less than 10 years [20,22,27], 10 to 20 years [9,10,24,28] and more than 20 years [12,21,23,25,26]. These study characteristics are summarized in Table 1.

### 3.3. Risk of Bias Assessment

Quality assessment by the NOS showed that the total scores for all the included studies ranged between five to eight. The scores for each domain ranged from two to four for the selected domain, zero to one for the comparability domain and three for the outcome domain (Table 2). None of these studies were excluded from the analysis due to the limited number of studies on this rare disease.

### 3.4. Results of Individual Study

The proportions of AVM hemorrhages per pregnancy ranged from 0.01 (95% CI: <0.001, 0.07) [27] to 0.89 (95% CI: 0.77, 0.96) [9]. Two studies [12,24] showed a large proportion with more than 30% of AVM hemorrhages per pregnancy (Table 3).

The pooled proportion of AVM hemorrhages per pregnancy after double arcsine transformation and converted back to the original proportion was 0.16 (95% CI: 0.08, 0.26). The heterogeneity chi-squared for this model was 296.58 (*p*-value < 0.001), and the percentage of variation across studies I^2^ was 96.29%, which is interpreted as considerable heterogeneity (Figure 2). However, without transformation, the pooled proportion of AVM hemorrhages per pregnancy was 0.18 (95% CI: 0.12, 0.24) with chi-squared of 458.88 (*p*-value < 0.001) and I^2^ of 97.60%.

### 3.5. Results of Subgroup Analyses

#### 3.5.1. Region

Subgroup analysis by world regions showed that the pooled proportion of AVM hemorrhages per pregnancy was highest in Europe [0.35 (95% CI: 0.02, 0.79)], followed by America [0.14 (95% CI: 0.04, 0.29)], and the lowest proportion was Asia [0.06 (95% CI: 0.01, 0.14)]. Heterogeneities for this subgroup were considerable, with 94% for America and 88% for Asia. However, the heterogeneity for Europe was not estimated due to the small number of studies. The heterogeneity of the region subgroup was not significantly different (*p*-value = 0.186) (Table 4, Figure 3).

#### 3.5.2. Study Design

Analysis of the subgroups by study designs showed that the retrospective review studies that pooled the proportion of AVM hemorrhages per pregnancy were 0.18 (95% CI: 007, 0.32). Meanwhile, the pooled proportion of case-cross-over design for the two studies was 0.05 (95% CI: 0.03, 0.06). Heterogeneity was estimated only for the retrospective review study of 96.6%, which was interpreted as considerable heterogeneity with a significant subgroup effect (*p*-value = 0.007) (Table 4, Figure 4).

#### 3.5.3. Duration of Study

The duration of the study was divided into three groups, and the subgroup analysis showed that the duration of the study from 10 to 20 years had the highest pooled proportion of AVM hemorrhages per pregnancy (0.37 (95% CI: 0.06, 0.77)). The pooled proportion for studies with more than 20 years was 0.10 (95% CI: 0.03, 0.21), and the pooled proportion for the studies with less than 10 years was the lowest at 0.05 (95% CI: 0.02, 0.10). Heterogeneity was considerable for all groups in this subgroup analysis, with I^2^ more than 90% for all groups except for the “less than 10 years” group, in which the heterogeneity was not estimable due to a limited number of studies. However, the heterogeneity between groups was not statistically significant (*p*-value = 0.074) (Table 4, Figure 5).

### 3.6. Publication Bias

The funnel plot of the pooled proportion of AVM hemorrhage per pregnancy showed asymmetry (Figure 6), and the Egger’s test indicated a small study effect (*p*-value = 0.024). However, in subgroup analyses, Egger’s test for all the subgroup analyses was not significant (*p*-value > 0.05), which was interpreted as having no small study effects except for the cross-over design. Egger’s test was not estimable because only two studies were involved in the design (Table 5).

### 3.7. Outlier and Sensitivity Analysis

Two studies were detected with large effect sizes [9,24]. Re-analyses with and without the studies by the leave-one-out method were performed to assess the influential effect on the pooled proportions. These sensitivity analyses showed that the studies were not very influential on the pooled proportion and the heterogeneities except for the subgroup analysis for the American region. It showed different large values for the pooled proportion and its heterogeneity with and without the study [24] (Table 6). Both studies were retained in the analysis since they did not affect the overall pooled proportion too much.

## 4. Discussion

This systematic review and meta-analysis determined that the risk of AVM hemorrhage was 16% (95% CI: 8%, 26%) per pregnancy in patients with AVM. A review [29] showed that 23% of intracranial hemorrhage during pregnancy was attributed to AVM. We did not manage to find other studies to compare the pooled proportion. A study in China [27] assessed the proportion of AVM hemorrhage per pregnancy in six studies published from 1990 to 2014 but did not pool the data analysis. However, a previous meta-analysis [28] reported the pooled odd ratio of the annual hemorrhage rate of patients with AVM between pregnancy (5.59%) and non-pregnancy (2.52%) of 3.19 (95% CI 1.52, 6.70) involving nine studies from the year 1990 to 2017. Our review did not estimate the annual hemorrhage rate due to incomplete data on the person–year and the unclear definition of the exposure period in some included studies.

The proportion of AVM hemorrhage per pregnancy in AVM patients indicated that the risk of hemorrhage during pregnancy was low in this review. A previous study [20] showed that the risk of hemorrhage remained at similar percentage rates in maternal-age women with pregnancy during the first trimester (5.3% per year) and third trimester (5.8% per year) or without pregnancy (4.5% per year). However, the risk of hemorrhage increased during the second trimester (17% per year). Contradiction to this result, another study [22] showed that the rate of hemorrhage was higher in pregnant women (10.8% per year) compared to non-pregnant women (1.1% per year) with a hazard ratio of 7.91 (95% CI: 2.64, 23.71). Meanwhile, a study in China [26] did not find any evidence that showed pregnancy increased the risk of AVM hemorrhage by the odds ratio of AVM hemorrhage during pregnancy compared to the non-pregnancy period of 0.71 (95% CI: 0.61, 0.82). With the limited number of studies available on the risk or rate of AVM hemorrhage, the evidence was limited and unclear to conclude that pregnancy may increase the risk of AVM hemorrhage.

Subgroup analyses were conducted to investigate the source of heterogeneity since the heterogeneity in this review was large. The subgroup analyses were conducted by a visual inspection of the forest plots for each subgroup. Of the three subgroup analyses, only the study design analysis showed a significant subgroup effect by heterogeneity. However, the evidence of this subgroup effect was ambiguous due to the inconsistency of the individual studies. The Cochrane handbook mentioned that the investigation of heterogeneity needs at least 10 studies to produce meaningful findings but is not adequate if the distribution of the studies is uneven, especially in a case of limited data available for the specific subgroup [30], as occurred in this review.

This review is not without limitations. A limited number of published studies on the epidemiological data of AVM hemorrhage during pregnancy narrowed the authors’ choice to select good-quality studies. Listed as a rare disease, there was a limited study on AVM, especially regarding incidence or prevalence. A previous study [31] stated that estimating the prevalence of rare diseases is challenging since epidemiological reports are frequently few. The research may be standardized or difficult to integrate, lack firmly established and specific diagnostic criteria and may be skewed based on the region studied. In addition, there are methodological difficulties unique to measuring small populations.

This review is based on available data. We included all studies regardless of the study quality in order to have a comprehensive range of data. There is a possibility of publication bias due to limited database access and language barrier.

## 5. Conclusions

The pooled proportion of AVM hemorrhage during pregnancy was low. However, the authors did not conclude whether the proportion was under- or overestimated due to a limited number of reviews in this area. Future epidemiological studies are needed to provide more information on AVM patients’ data, treatment and management. No doubt, it is challenging in terms of methodology and related resources to conduct a study on a rare disease, especially during pregnancy.

## Figures and Tables

**Figure 1 ijerph-19-13183-f001:**
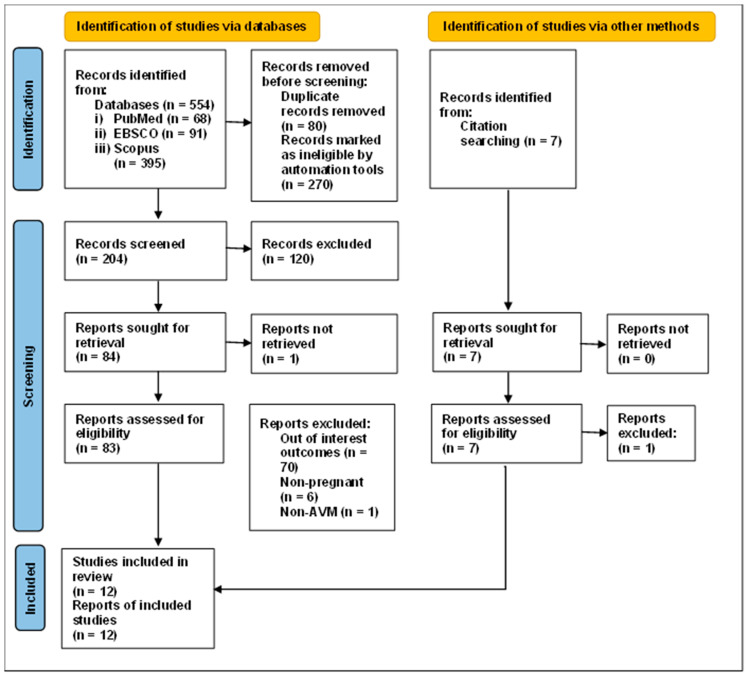
PRISMA 2020 flow chart.

**Figure 2 ijerph-19-13183-f002:**
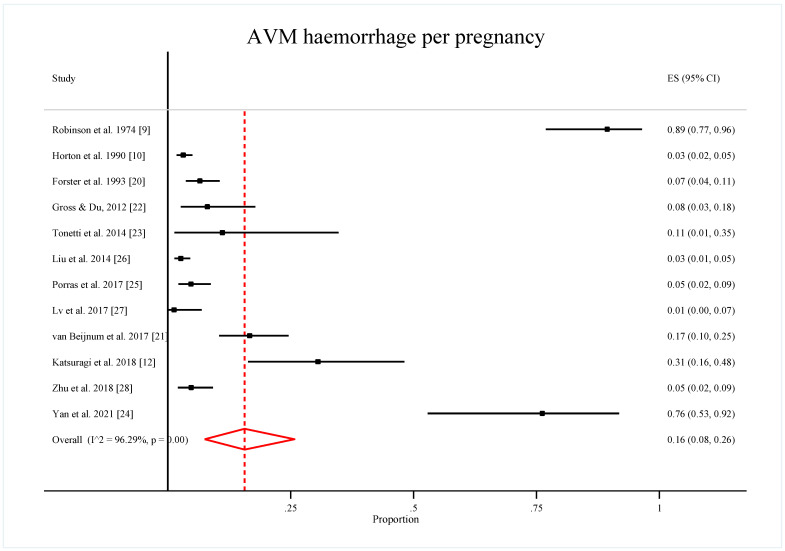
Forest plot of the pooled proportion of AVM hemorrhage per pregnancy by double arcsine transformation method [9,10,12,20,21,22,23,24,25,26,27,28].

**Figure 3 ijerph-19-13183-f003:**
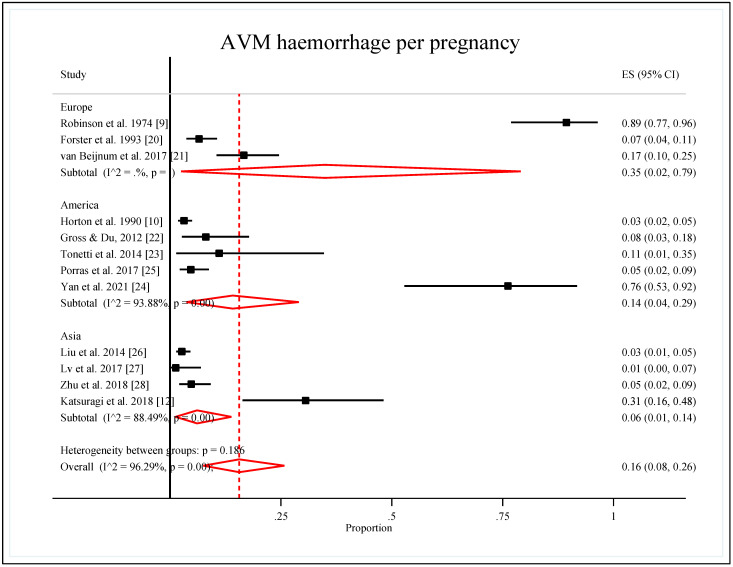
Forest plot of the pooled proportion of AVM hemorrhage per pregnancy by regions [9,10,12,20,21,22,23,24,25,26,27,28].

**Figure 4 ijerph-19-13183-f004:**
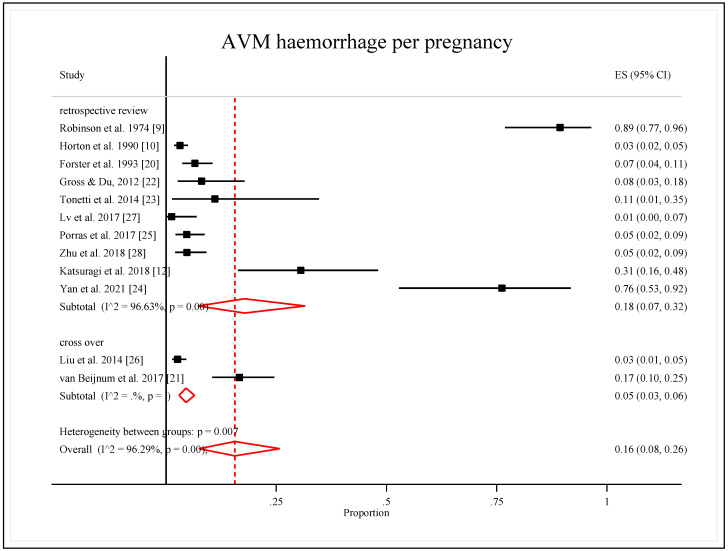
Forest plot of the pooled proportion of AVM hemorrhage per pregnancy by study design [9,10,12,20,21,22,23,24,25,26,27,28].

**Figure 5 ijerph-19-13183-f005:**
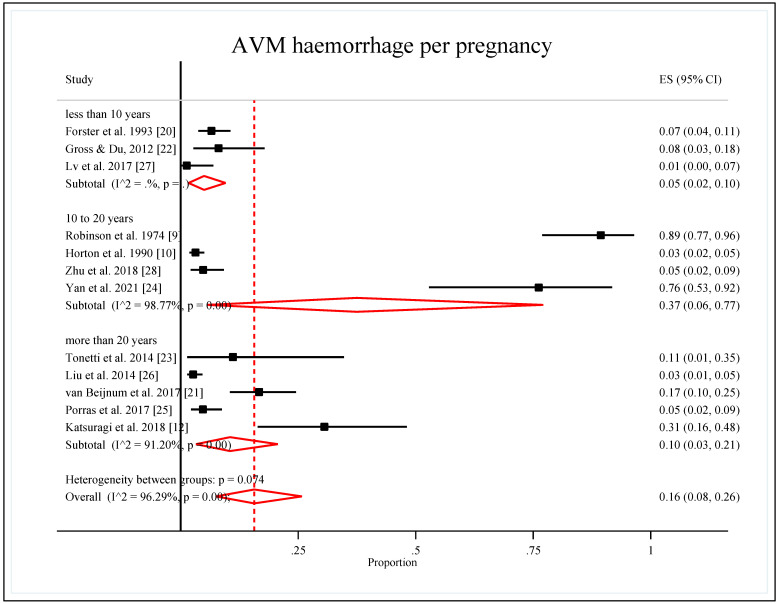
Forest plot of the pooled proportion of AVM hemorrhage per pregnancy by study duration [9,10,12,20,21,22,23,24,25,26,27,28].

**Figure 6 ijerph-19-13183-f006:**
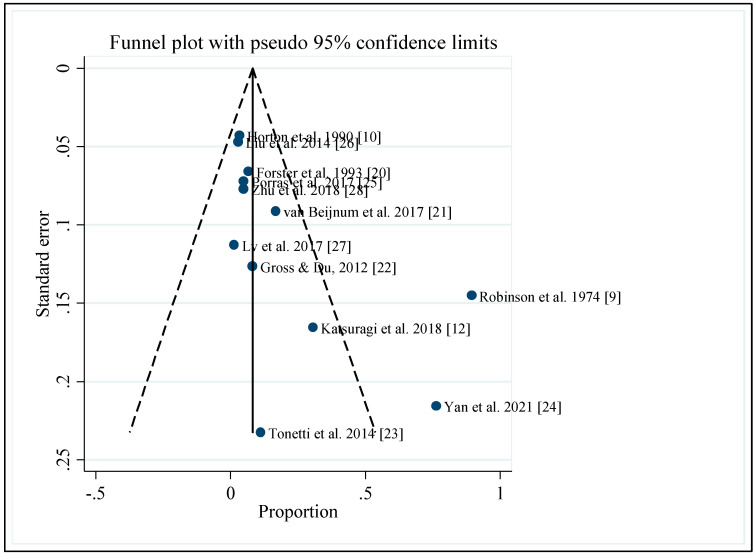
Funnel plot for the pooled proportion of AVM hemorrhage per pregnancy [9,10,12,20,21,22,23,24,25,26,27,28].

**Table 1 ijerph-19-13183-t001:** Study characteristics of included studies (n = 12).

Study	Country	Study Duration	Study Design	Number of Patients	No. Pregnancies	Hemorrhages during Pregnancy
Forster	UK	1985–1991	Retrospective review	191	229	15
Gross	US	2002–2010	Retrospective review	58	62	5
Horton	US	1977–1986	Retrospective review	451	540	17
Katsuragi	Japan	1981–2013	Retrospective review	26	36	11
Liu	China	1960–2010	Case-cross-over	393	452	12
Lv	China	2006–2010	Retrospective review	67	78	1
Porras	North America	1990–2015	Retrospective review	270	191	9
Robinson	UK	1954–1970	Retrospective review	146	47	42
Tonetti	US	1987–2012	Retrospective review	253	18	2
van Beijnum	Scotland	1990–2010	Case-cross-over and self-controlled case-series	139	120	20
Yan	US	2000–2017	Retrospective review	320	21	16
Zhu	China	2006–2017	Retrospective review	264	168	8

**Table 2 ijerph-19-13183-t002:** Risk of bias assessment by the Newcastle Ottawa Scale of included studies (n = 12).

Author	Year	Selection	Representativeness of the Exposed Cohort	Selection of the Non-Exposed Cohort	Ascertainment of Exposure	Demonstration that Outcome of Interest Was Not Present at Start of Study	Comparability	Comparability of Cases and Controls on the Basis of the Design or Analysis	Outcomes	Assessment of Outcome	Was Follow-Up Long Enough for Outcomes to Occur	Adequacy of Follow Up of Cohorts	Total Score
Robinson	1974	3	*	*	*		1	*	3	*	*	*	7
Horton	1990	3	*	*	*		1	*	3	*	*	*	7
Forster	1993	3	*	*	*		1	*	3	*	*	*	7
Gross	2012	3	*	*	*		1	*	3	*	*	*	7
Tonetti	2014	3	*	*	*		1	*	3	*	*	*	7
Liu	2014	4	*	*	*	*	1	*	3	*	*	*	8
Lv	2017	3	*	*	*		1	*	3	*	*	*	7
van Beijnum	2017	4	*	*	*	*	1	*	3	*	*	*	8
Porras	2017	3	*	*	*		1	*	3	*	*	*	7
Zhu	2018	3	*	*	*		1	*	3	*	*	*	7
Katsuragi	2018	2	*		*		0		3	*	*	*	5
Yan	2021	2	*		*		0		3	*	*	*	5

* stars for each item in the selection, comparability and outcomes domains.

**Table 3 ijerph-19-13183-t003:** The outcomes of individual study.

Author	Year	Proportion (95% Confidence Interval)	Standard Error	Random Weight
Robinson	1974	0.89 (0.77, 0.96)	0.15	8.11
Horton	1990	0.03 (0.02, 0.05)	0.04	8.99
Forster	1993	0.07 (0.04, 0.11)	0.07	8.87
Gross	2012	0.08 (0.03, 0.18)	0.13	8.32
Tonetti	2014	0.11 (0.01, 0.35)	0.23	6.94
Liu	2014	0.03 (0.01, 0.05)	0.05	8.98
Lv	2017	0.01 (<0.001, 0.07)	0.09	8.68
van Beijnum	2017	0.17 (0.10, 0.25)	0.07	8.83
Porras	2017	0.05 (0.02, 0.09)	0.11	8.47
Zhu	2018	0.05 (0.02, 0.09)	0.08	8.79
Katsuragi	2018	0.31 (0.16, 0.48)	0.17	7.85
Yan	2021	0.76 (0.53, 0.92)	0.22	7.17

**Table 4 ijerph-19-13183-t004:** Results of subgroups analyses.

Subgroup	Studies	Proportion (95% CI)	I^2^ (%)	*p*-Value
Region				
Europe	3	0.35 (0.02, 0.79)	NA	NA
America	5	0.14 (0.04, 0.29)	93.88	<0.001
Asia	4	0.06 (0.01, 0.14)	88.49	<0.001
Study design				
Retrospective review	10	0.18 (0.07, 0.32)	96.63	<0.001
Case cross-over	2	0.05 (0.03, 0.06)	NA	NA
Study duration				
<10 years	4	0.05 (0.02, 0.10)	NA	NA
10 to 20 years	3	0.37 (0.06, 0.77)	98.77	<0.001
>20 years	5	0.10 (0.03, 0.21)	91.20	<0.001

NA = Not available.

**Table 5 ijerph-19-13183-t005:** Egger’s test of the study.

Egger Test	Coefficient	(95% CI)	*p*-Value
Overall	2.76	(0.45, 5.07)	0.024
Region			
Europe	9.86	(−26.40, 46.13)	0.179
America	1.87	(−1.74, 5.48)	0.198
Asia	1.38	(−2.78, 5.54)	0.289
Study design			
Retrospective review	2.77	(−0.08, 5.63)	0.055
Case-cross over	3.18	NA	NA
Study duration			
<10 years	−0.29	(−11.17, 10.59)	0.793
10 to 20 years	5.52	(−3.69, 14.72)	0.123
>20 years	1.46	(−0.84, 3.76)	0.136

NA = Not available.

**Table 6 ijerph-19-13183-t006:** Sensitivity analysis for outlier studies.

	With Outliers		Without Robinson, Hall and Sedzimir [9]		Without Yan, Ko, Hetts, Weinsheimer, Abla, Lawton and Kim [24]	
	Pooled Proportion (95% CI)	I^2^	Pooled Proportion (95% CI)	I^2^	Pooled Proportion (95% CI)	I^2^
Overall	0.16 (0.08, 0.26)	96.29	0.10 (0.05, 0.15)	91.19	0.12 (0.05, 0.21)	95.87
Region						
Europe	0.35 (0.02, 0.79)	NA	0.10 (0.07, 0.13)	NA	0.35 (0.02, 0.79)	NA
America	0.14 (0.04, 0.29)	93.88	0.14 (0.04, 0.29)	93.88	0.04 (0.02, 0.07)	48.19
Asia	0.06 (0.01, 0.14)	88.49	0.06 (0.01, 0.14)	88.49	0.06 (0.01, 0.14)	88.49
Study design						
Retrospective review	0.18 (0.07, 0.32)	96.63	0.10 (0.05, 0.17)	90.9	0.13 (0.05, 0.25)	96.27
Case-cross over	0.05 (0.03, 0.06)	NA	0.05 (0.03, 0.06)	NA	0.05 (0.03, 0.06)	NA
Study duration						
<10 years	0.05 (0.02, 0.10)	NA	0.05 (0.02, 0.10)	NA	0.05 (0.02, 0.10)	NA
10 to 20 years	0.37 (0.06, 0.77)	98.77	0.18 (0.03, 0.43)	NA	0.26 (0.01, 0.68)	NA
>20 years	0.10 (0.03, 0.21)	91.20	0.10 (0.03, 0.21)	91.2	0.10 (0.03, 0.21)	91.2

NA = Not available.

## Data Availability

The data are contained within the article.

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
