# Peer review of "Arteriovenous Malformation Hemorrhage in Pregnancy: A Systematic Review and Meta-Analysis"

_ijerph, 2022, doi:10.3390/ijerph192013183_

Round 1

Reviewer 1 Report

Summary

The review by Yusof et al. seeks to determine if pregnancy increases the risk for hemorrhages in patients with AVMs. The authors performed a combined analysis of previous studies on this topic. This work is very important.

Major comments

1) The authors describe AVMs as congenital vascular lesions. However, AVMs can also grow later in life.

2) AVMs can be sporadic or associated with an inherited disease, such as Hereditary Hemorrhagic Telangiectasia. This review would greatly benefit from elaborating on this. If at all possible, it would be very interesting to repeat the subgroup analysis based on these two distinct etiologies.

Minor comment

I would strongly recommend English language editing, with the main focus on improving clarity.

Author Response

Kindly refer to the attachment. Thank you

Reviewer 2 Report

Ruhana Che Yusof and colleagues made a meta-analysis of AVM hemorrhage in pregnancy, and the results showed that AVM hemorrhage in per pregnancy was considered to be low. This research is meaningful. However, there are some problems:

1.      The included literature are relatively few, with only 12 articles;

2.      There are only four countries involved in the region (the United States, Britain, Japan and China);

3.      The span of years for inclusion articles is too large, from 1974 to 2021;

4.      Samples of patients included in some studies and pregnant women differ too much;

5.      A literature study spans 50 years. How can we ensure the reliability of the results?

6.      The heterogeneity of the article is too great, and subgroup analysis is not fully discussed.

Author Response

(The authors gave the same response as above.)
